# Muscimol inhibits plasma membrane rupture and ninjurin-1(NINJ1) oligomerization during pyroptosis

Andreas B. den Hartigh[1], Wendy P. Loomis[1], Marisa J. Anderson[1], Bente Frølund[2] & Susan L. Fink [1]✉

Pyroptosis is a cell death process that causes inflammation and contributes to numerous diseases. Pyroptosis is mediated by caspase-1 family proteases that cleave the pore-forming protein gasdermin D, causing plasma membrane rupture and release of pathogenic cellular contents. We previously identified muscimol as a small molecule that prevents plasma membrane rupture during pyroptosis via an unidentified mechanism. Here, we show that muscimol has reversible activity to prevent cellular lysis without affecting earlier pyroptotic events. Although muscimol is a well-characterized agonist for neuronal GABA$_A$ receptors, muscimol protection is not altered by GABA$_A$ receptor antagonists or recapitulated by other GABA$_A$ agonists, suggesting that muscimol acts via a novel mechanism. We find that muscimol blocks oligomerization of ninjurin-1, which is required for plasma membrane rupture downstream of gasdermin D pore formation. Our structure-activity relationship studies reveal distinct molecular determinants defining inhibition of pyroptotic lysis compared to GABA$_A$ binding. In addition, we demonstrate that muscimol reduces lethality during LPS-induced septic shock. Together, these findings demonstrate that ninjurin-1-mediated plasma membrane rupture can be pharmacologically modulated and pave the way toward identification of therapeutic strategies for pathologic conditions associated with pyroptosis.

[1] Department of Laboratory Medicine and Pathology, University of Washington, Seattle, WA, USA. [2] Department of Drug Design and Pharmacology, Faculty of Health and Medical Sciences, University of Copenhagen, Copenhagen, Denmark. ✉email: sfink@uw.edu

Pyroptosis is a regulated process of cell death that protects against infection[1], but also contributes to the pathogenesis of leading global causes of mortality, including cardiovascular disease[2,3], stroke[4,5], sepsis[6–8], and neurodegeneration[9,10]. Pyroptosis was initially defined by dependence on proteases in the caspase-1 family, which are activated by sensing of microbial or damage-associated stimuli via inflammasomes[11]. Active caspase-1 cleaves gasdermin D, releasing the N-terminal pore-forming domain and leading to plasma membrane rupture; thus, a revised definition of pyroptosis is gasdermin-dependent lytic cell death[12]. Pore formation during pyroptosis was previously thought to cause plasma membrane rupture solely as a result of osmotic forces[13]. Recently published data demonstrate that the homophilic adhesion molecule ninjurin-1 oligomerizes during pyroptosis and is required for plasma membrane rupture downstream of gasdermin D pore formation[14].

Plasma membrane rupture, or cell lysis, releases intracellular contents that contribute to local and systemic pathology[15,16]. For example, the chromatin-associated protein HMGB1 plays an extracellular role in inflammation and is released from pyroptotic cells via a gasdermin D and ninjurin-1-dependent process[14,17]. In addition, oligomerization of the inflammasome adapter ASC forms visible "specks" that are released from pyroptotic cells to propagate inflammation[18] and seed other protein aggregates, such as amyloid-β plaques in Alzheimer's disease[19]. Pyroptotic macrophages also release clotting factors causing disseminated intravascular coagulation, multi-organ failure, and lethality[16]. This process occurs in settings such as sepsis, where infection triggers pyroptosis[20].

Strategies to inhibit inflammasome and caspase-1 activation have been widely pursued based on the role of these processes in diverse human diseases[21,22]. However, it is increasingly appreciated that cell death pathways are redundant and interconnected. For example, caspase-1 inactivation does not prevent inflammasome-triggered cell death but redirects cells to an alternative pathway of lytic death[23]. Thus, preventing the final stage of plasma membrane rupture could be an alternative approach to limit the release of pathogenic factors in a variety of human diseases. We previously identified muscimol as a novel small molecule that prevents plasma membrane rupture during pyroptosis[24]. In this study, we further explored the mechanism by which muscimol prevents cellular lysis and identified the structure-activity relationship for the inhibition of lysis. In addition, we demonstrate that muscimol reduces lethality during lipopolysaccharide (LPS)-induced septic shock.

## Results

**Muscimol prevents plasma membrane rupture without affecting earlier pyroptotic events.** *Salmonella* infection and anthrax lethal toxin treatment trigger pyroptosis in murine bone marrow-derived macrophages via the NLRC4 and NLRP1b inflammasomes, respectively[25]. Consistent with our prior study[24], we observed that muscimol prevents plasma membrane rupture from *Salmonella*-infected and lethal toxin-treated pyroptotic macrophages, as assessed by measuring release of the large tetrameric cytoplasmic enzyme, lactate dehydrogenase (LDH) (Fig. 1a). In addition, we observed that muscimol inhibits LDH release in response to NLRP3 inflammasome activation with nigericin and post-apoptotic lysis induced by staurosporine (Fig. 1a). Since plasma membrane rupture is the final event in the pyroptotic cascade, we sought to determine whether muscimol affects the upstream events that precede lysis. During inflammasome activation, the adapter protein ASC oligomerizes to form a single large speck or focus per cell, which can be visualized in primary macrophages expressing ASC tagged with a fluorescent protein[26].

To determine if muscimol affects *Salmonella*-induced inflammasome activation, we infected ASC-Citrine-expressing macrophages and monitored ASC foci formation using live cell microscopy. In mock-infected macrophages treated with phosphate-buffered saline (PBS), the ASC-Citrine fusion protein is present diffusely throughout the cell, whereas redistribution to a single focus is observed in the majority of *Salmonella*-infected cells (Fig. 1b). We found that formation of ASC foci in response to *Salmonella* was unaffected by muscimol (Fig. 1b, c), suggesting that muscimol does not inhibit lysis by altering upstream inflammasome activation. This is consistent with our previous finding that muscimol does not affect either *Salmonella* or lethal toxin-induced caspase-1 activation, as assessed by labeling with the caspase-1 activity probe FAM-YVAD-FMK[24].

Once activated, caspase-1 cleaves gasdermin D, separating the N-terminal pore-forming domain from the C-terminal repressor domain. To determine whether muscimol affects gasdermin D proteolysis, we assessed gasdermin D in *Salmonella*-infected cells by Western blot and found that cleavage was unaffected by muscimol (Fig. 1d). The pore-forming domain of gasdermin D binds lipids on the plasma membrane inner leaflet and oligomerizes to form pores which allow the influx of small molecules such as the membrane-impermeable fluorescent nucleic acid stain, TO-PRO-3[25]. Kinetic analysis of TO-PRO-3 uptake in wild type (WT), gasdermin D-deficient, or caspase-1/11-deficient macrophages confirmed that gasdermin D and caspase-1 are required for TO-PRO-3 uptake in response to *Salmonella* infection (Fig. 1e and Sup Fig. 1a). Importantly, muscimol had no effect on TO-PRO-3 uptake in *Salmonella*-infected WT macrophages, indicating that gasdermin D pore formation was unaffected. Gasdermin D pores are sufficient to mediate loss of cytosolic ATP and cessation of metabolic activity, indicating cell death, independently of plasma membrane rupture[27,28]. We found reduced cellular ATP content in *Salmonella*-infected cells compared to uninfected controls, which was not rescued by muscimol (Sup Fig. 1b). These results suggest that muscimol does not affect gasdermin D cleavage, pore formation, or consequent cessation of metabolic activity.

Interleukin (IL)−1β is an inflammatory cytokine that is synthesized as a precursor, requiring proteolytic activation by enzymes including caspase-1. Release of processed IL-1β through gasdermin D pores is facilitated by a charge-dependent mechanism involving electrostatic filtering[29]. We quantified IL-1β in the supernatant of pyroptotic macrophages and found similar IL-1β secretion by *Salmonella*-infected macrophages in the presence or absence of muscimol (Fig. 2a), confirming that muscimol does not prevent gasdermin D pore formation or cargo release. HMGB1 is another cellular protein released following inflammasome activation. Although gasdermin D pores are sufficient to allow secretion of activated IL-1β, HMGB1 release requires ninjurin-1-mediated plasma membrane rupture[14]. To determine if muscimol prevents HMGB1 release from pyroptotic macrophages, we assessed HMGB1 levels in the supernatant by western blot (Fig. 2b, c). We found that release of HMGB1 from pyroptotic macrophages was reduced by muscimol (Fig. 2b, c). The cytoprotectant glycine also prevents plasma membrane rupture during pyroptosis without altering pyroptotic pore formation[13] and similarly reduces HMGB1 release (Fig. 2b, c). Together, these results indicate that muscimol prevents plasma membrane rupture without affecting earlier pyroptotic events.

**Muscimol protects against LPS-induced sepsis.** Pyroptosis plays a critical role in the pathogenesis of sepsis, and mice unable to

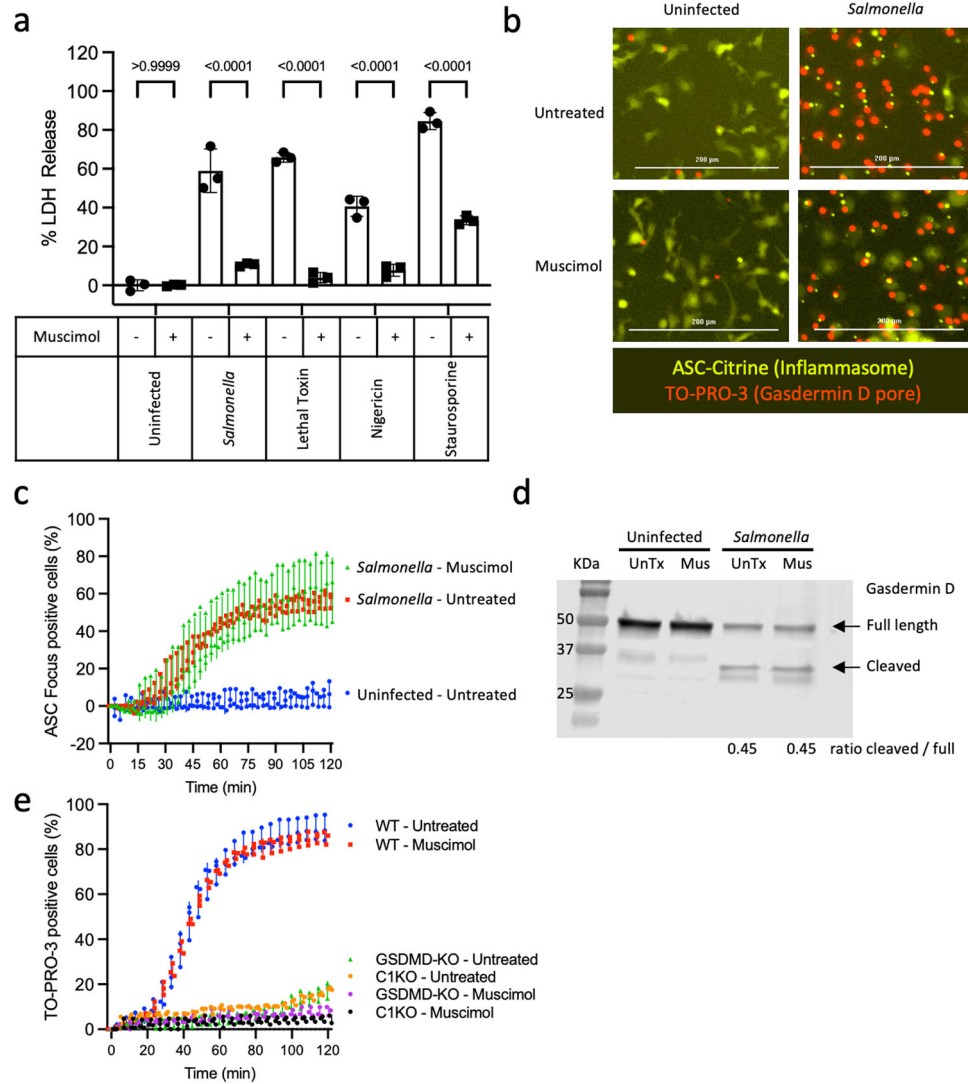

**Fig. 1 Muscimol prevents pyroptotic lysis without affecting the formation of ASC inflammasome foci or membrane pores. a** Bone marrow-derived macrophages (BMMs) were treated with *Salmonella*, Lethal Toxin, Nigericin, or Staurosporine in the presence of 1 mM muscimol as indicated. LDH released during cell lysis was measured. **b** Localization of the inflammasome adapter ASC (yellow) and uptake of the small membrane-impermeant nuclear dye TO-PRO-3 (red) were assessed in ASC-Citrine-expressing BMMs infected with *Salmonella* for 90 min. **c** ASC foci formation was quantified. **d** Cleavage of gasdermin D was determined by Western blot in untreated (UnTx) and muscimol (Mus) treated cells infected with *Salmonella* or uninfected controls. The ratio of cleaved/uncleaved gasdermin D was determined. **e** TO-PRO-3 uptake was assessed in wild type, caspase$-1/11^{-/-}$, or gasdermin $D^{-/-}$ BMMs infected with *Salmonella* in the presence or absence of muscimol. The percentage of cells with TO-PRO-3+ nuclei was quantified. Representative data in **a**, **c**, and **e** (mean ± SD, n = 3) from two or three independent experiments are shown. Images representative of 5 taken per condition in three independent experiments are shown in b; scale bars represent 200 mm. Western blot representative of two independent experiments is shown in **d**. Statistics: 2-way ANOVA + Tukey's multiple comparisons.

undergo caspase-11 or gasdermin D-dependent pyroptosis are protected from LPS-induced lethality[30–32]. To determine if muscimol treatment provides a protective effect in vivo, we examined LPS-induced sepsis in mice. We treated mice with muscimol subcutaneously 30 minutes before intraperitoneal challenge with LPS and observed that a single dose of muscimol was able to protect mice from LPS-induced septic shock (Fig. 3a). Organ damage is a major contributor to lethality in sepsis, and kidney failure is a particularly common complication[33]. We measured blood urea nitrogen (BUN), which is assessed clinically in the evaluation of renal function, and found markedly elevated BUN levels in LPS-treated mice, consistent with sepsis-associated renal failure (Fig. 3b). Remarkably, the majority of LPS-treated mice that received muscimol had normal BUN levels (Fig. 3b), suggesting protection from renal failure.

**Muscimol protection against plasma membrane rupture is reversible**. To further understand the mechanism by which muscimol blocks pyroptotic cell lysis, we next investigated the durability of muscimol inhibition. Muscimol is a classic agonist for neuronal GABA$_A$ receptors and acts by reversibly binding the orthosteric ligand binding site[34]. To determine if the ability of muscimol to protect cells from lysis is reversible, we performed sequential measurements of LDH released from *Salmonella*-infected macrophages in the presence or absence of muscimol (Fig. 4a). After 60 min of infection, we collected the supernatant (LDH1, Fig. 4b), washed the cells with warm cell culture medium and incubated for an additional 60 min in the presence or absence of muscimol (LDH2, Fig. 4c). When muscimol was removed, we observed release of LDH from *Salmonella*-infected macrophages that were previously protected by muscimol

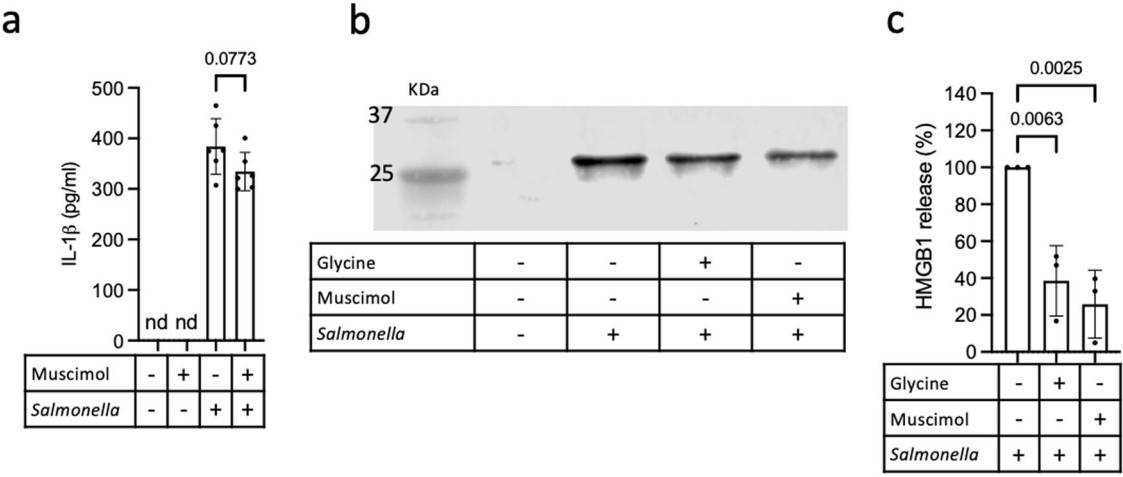

**Fig. 2 Muscimol prevents lytic release of inflammatory mediator HMGB. a** BMMs were infected with *Salmonella* or treated with PBS in the presence of muscimol as indicated and IL-1β secretion was measured by ELISA. Representative data (mean ± SD, $n = 6$) from three independent experiments are shown. **b**, **c** BMM were infected with *Salmonella* or treated with PBS in the presence of glycine or muscimol as indicated. Release of HMGB1 was determined by western blot (**b**) and quantified (**c**). Representative blot is shown in b and quantification in **c** is mean ± SD of $n = 3$ independent experiments. M: MW marker in KDa. Statistics: One-way ANOVA + Tukey's multiple comparisons. nd is none detected.

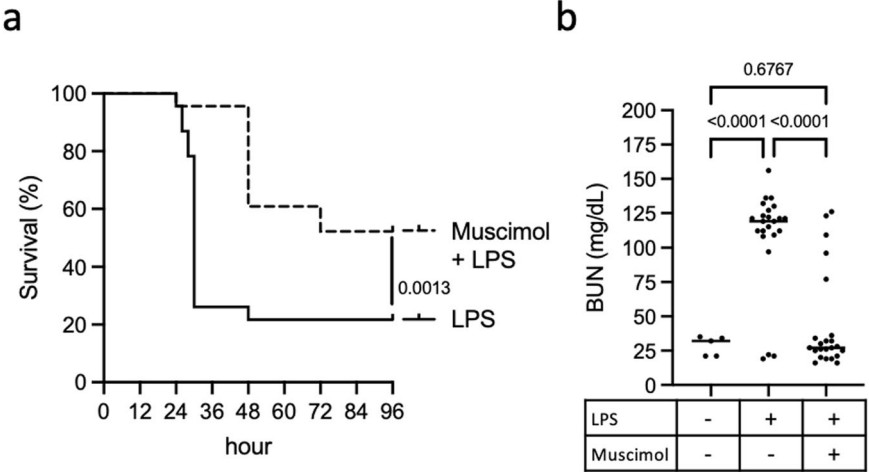

**Fig. 3 Muscimol protects mice against LPS-induced lethality. a** C57BL/6J mice were injected with 3 mg/kg muscimol, or PBS, followed by 10 mg/kg LPS. Mice were observed for clinical signs of illness and euthanized when overt signs of sepsis were observed. **b** Sera from mice injected with PBS, LPS alone, or muscimol + LPS were analyzed for the level of blood urea nitrogen (BUN). Data are combined values from three independent experiments, $n = 23$ mice per condition for LPS and muscimol + LPS, $n = 5$ uninfected mice. Statistics: (**a**) Mantel-Cox Log-rank test, (**b**) one-way ANOVA + Tukey's multiple comparisons.

(Fig. 4c, media + muscimol → media). However, the continued presence of muscimol provided persistent protection against LDH release (Fig. 4c, media + muscimol → media + muscimol). No further LDH release was observed in the second hour from cells that were not initially protected by muscimol (Fig. 4c, media → media, or media → media + muscimol) since all the susceptible cells underwent pyroptotic lysis during the first hour (LDH1, Fig. 4b). Together, these findings suggest that muscimol protection against plasma membrane rupture during pyroptosis is a reversible phenomenon.

**GABA_A receptor modulators do not alter muscimol protection.** Ionotropic GABA_A receptors are heteropentameric structures composed of many possible subunits and the orthosteric muscimol binding site is at the interface between α and β subunits[34]. We first sought to examine which (if any) GABA_A receptor subunits are expressed in differentiated murine bone marrow-derived macrophages. The relative expression of GABA_A receptor

subunits in bone marrow-derived macrophages and brain (positive control) of C57BL/6 and BALB/c mice was determined by RT-PCR. We found that most, but not all, subunits are expressed in bone marrow-derived macrophages (Sup Fig. 2), indicating that GABA_A receptors may potentially be present in these cells.

We previously found that GABA, the physiologic GABA_A receptor agonist, had no cytoprotective effect by itself[24], and additionally, we found that GABA does not interfere with the ability of muscimol to protect against pyroptotic lysis (Sup Fig. 3). Gabazine is a classic orthosteric antagonist of GABA_A receptors and competes with GABA for ligand binding[34]. We found that gabazine neither antagonized nor recapitulated the ability of muscimol to prevent pyroptotic plasma membrane rupture, as assessed by LDH release (Sup Fig. 3). Picrotoxin is another classic GABA_A receptor antagonist and acts to inhibit ligand-induced ion conductance via a non-competitive mechanism[34]. We found that picrotoxin also had no effect to antagonize or recapitulate muscimol's effect on pyroptosis (Sup Fig. 3). As the protective

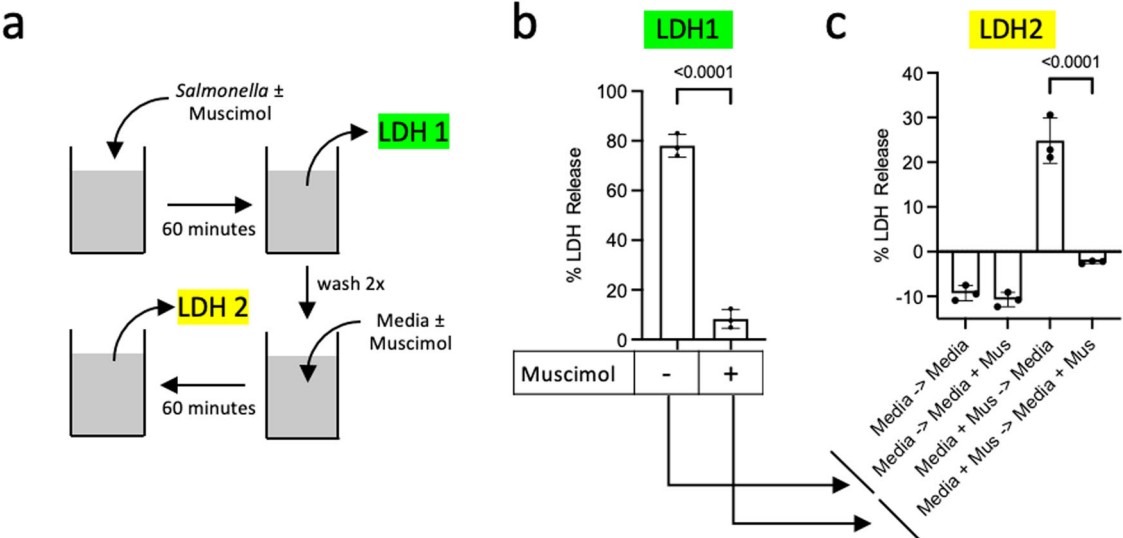

**Fig. 4 Inhibition of cellular lysis by muscimol is reversible. a** Schematic representation of the experiment. **b** BMM were infected with *Salmonella* for 60 min in a medium containing muscimol (as indicated) and cell lysis was measured by LDH release (LDH 1). **c** New medium with or without muscimol was added and LDH released during the next 60 min was measured (LDH 2). Statistics: *t*-test (**b**) or one-way ANOVA + Tukey's multiple comparisons (**c**). nd is none detected. Representative data in **b** and **c** (mean ± SD, $n = 3$) from two independent experiments are shown.

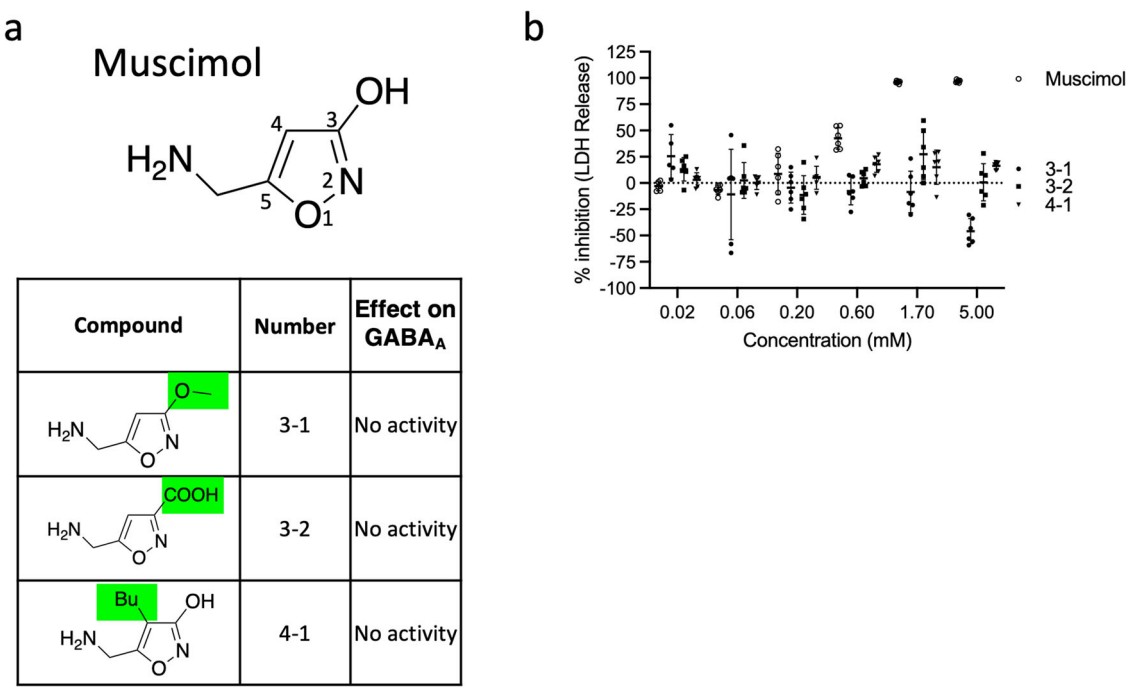

**Fig. 5 Modification of muscimol at positions 3 and 4 abolishes the inhibitory capacity of muscimol. a** Structures of muscimol (with numbers indicating positions of modification sites) and analogs modified at positions 3 and 4 of the ring structure. The effect on GABA$_A$ is based on previously published findings. **b** BMM were infected with *Salmonella* in the presence of indicated concentrations of muscimol or muscimol analogs. LDH released during cell lysis was measured and used to calculate the inhibitory percentage. Combined data from two independent experiments (mean ± SD, $n = 3$ per experiment) are shown.

effect of muscimol is not antagonized by either gabazine or picrotoxin nor recapitulated by GABA itself, together these results suggest that GABA$_A$ receptors may not be involved in inhibition of plasma membrane rupture.

**Structure-activity relationship of muscimol analogs**. To better understand the molecular requirements for muscimol inhibition,

we systematically tested a panel of muscimol analogs for their ability to prevent plasma membrane rupture during pyroptosis. Muscimol (5-aminomethyl-3-isoxazolol) is an aromatic heterocyclic compound with a five-membered 3-hydroxyisoxazole ring (Fig. 5a). Structure-activity studies of a number of muscimol analogs for GABA$_A$ receptor binding and agonist activity have defined the critical molecular determinants of ligand potency and revealed that some minor structural modifications result in

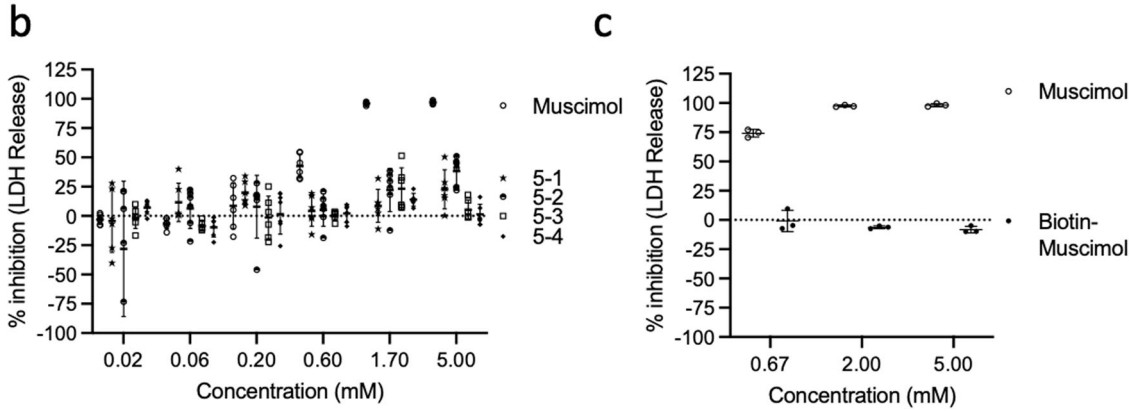

**Fig. 6 Modification of muscimol at position 5 abolishes the inhibitory capacity of muscimol. a** Structures of muscimol analogs modified at position 5 of the ring structure that maintains activity at the GABA_A receptor. **b, c** BMM were infected with *Salmonella* in the presence of indicated concentrations of muscimol or muscimol analogs. LDH released during cell lysis was measured and used to calculate the inhibitory percentage. Combined data from two independent experiments (mean ± SD, $n = 3$ per experiment) are shown.

dramatic loss of affinity for GABA_A receptors, whereas others have minimal effect[35]. We sought to determine whether distinct molecular determinants define inhibition of pyroptotic lysis.

First, we examined the substitution of the alcohol at position 3 of the ring structure with either methoxy (compound 3-1) or carboxylic acid (compound 3-2) groups and found that these changes completely abolished the protective effect against pyroptotic lysis (Fig. 5b). Addition of a butyl group at position 4 of the ring structure (compound 4-1) also completely eliminated inhibition of plasma membrane rupture (Fig. 5b). All of these substitutions also abolish GABA_A agonist activity[36].

Next, we tested substituents of the side chain at position 5, some of which retain GABA_A receptor binding and agonist activity[37]. The bicyclic muscimol analog 4-PIOL (compound 5-1)

retains partial GABA_A agonist activity[38], but demonstrated no inhibition of plasma membrane rupture (Fig. 6a, b). Likewise, analogs carrying insertions of a single carbon alkyl group at different positions in the side chain (compounds 5-2, 5-3, and 5-4) retain partial or weak GABA_A agonist activity[37,39], but failed to protect against pyroptotic lysis (Fig. 6a, b). Addition of a linker and terminal biotin results in substantial GABA_A agonist activity[40], but no inhibition of plasma membrane rupture (Fig. 6a, c). Further muscimol analogs with more complex side chain modifications (compounds 5-5 through 5-9) lose GABA_A receptor binding, as well as activity against pyroptosis (Sup Fig. 4a, b).

Finally, we examined a group of muscimol analogs that each contain substitutions at multiple positions (Fig. 7a). Most of these

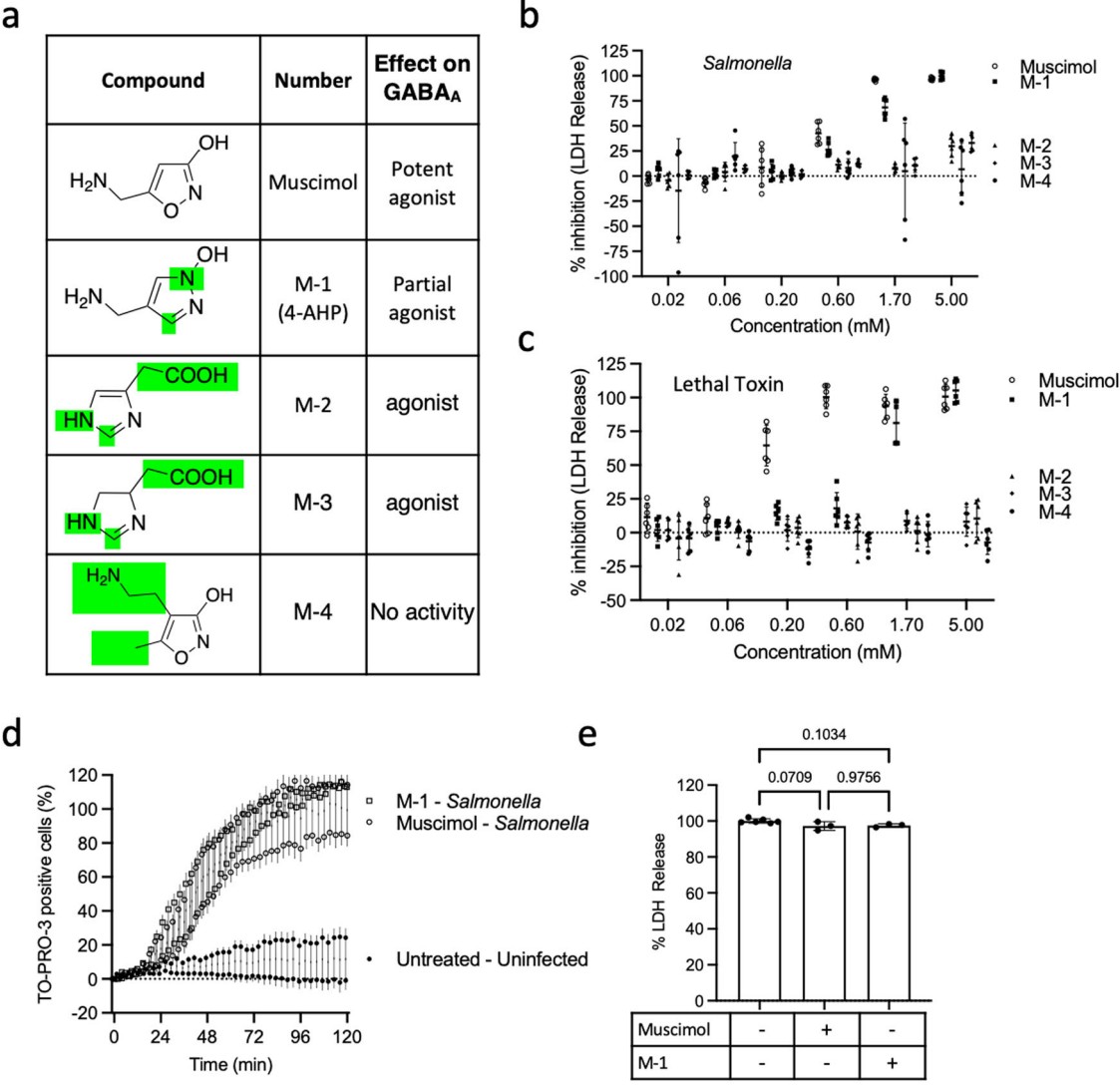

**Fig. 7 A muscimol analog with modifications at multiple positions retains inhibitory potential. a** Structures of muscimol analogs modified at multiple positions of the ring structure. **b**, **c** BMM were infected with *Salmonella* or exposed to Lethal Toxin in the presence of indicated concentrations of muscimol or muscimol analogs. LDH released during cell lysis was measured and used to calculate the inhibitory percentage. **d** TO-PRO-3 uptake was assessed in BMM infected with *Salmonella* in the presence of 1 mM muscimol or analog M1. **e** Total cell lysis was assessed in the presence of muscimol or analog M-1 to determine if the LDH assay was affected by the addition of muscimol or muscimol analogs. Data in **b**, **c**, and **d** are from two independent experiments (mean ± SD, $n = 3$ per experiment). Data in **e** are representative (mean ± SD, $n = 3$) from two independent experiments.

analogs (compounds M-2 through M-4) demonstrated no inhibition of plasma membrane rupture, even though some retain GABA$_A$ agonist activity (Fig. 7a–c). Notably, we discovered that one compound (M-1), the bioisosteric 4-(aminomethyl)−1-hydroxypyrazole (4-AHP) analog[41] inhibited LDH release from *Salmonella*-infected macrophages, with a modest reduction in potency compared to muscimol (Fig. 7b). This effect was not limited to *Salmonella*-induced, NLRC4-mediated pyroptosis, as we observed similar inhibition in macrophages treated with anthrax lethal toxin, which activates NLRP1b (Fig. 7c). This compound had no effect on TO-PRO-3 uptake, indicating that gasdermin D pore formation was unaltered (Fig. 7d), nor did it have an effect on detection of LDH from detergent-treated cells (Fig. 7e). Together, these results demonstrate that most muscimol structural modifications abolish protection against pyroptotic lysis, including some that have minimal effect on GABA$_A$ receptor binding and agonist activity. Therefore, remarkably constrained molecular determinants define inhibition of pyroptotic lysis and this unique structure-activity relationship further supports the hypothesis that muscimol activity is via a novel, GABA receptor-independent mechanism.

**Muscimol protects against lysis by preventing ninjurin-1 oligomerization.** The finding that ninjurin-1 is required for plasma membrane rupture downstream of gasdermin D pore formation suggested the hypothesis that muscimol may target ninjurin-1-mediated events to prevent plasma membrane rupture. We confirmed that LDH release was significantly reduced from *Salmonella*-infected ninjurin-1-deficient macrophages compared to cells from ninjurin$^{+/+}$ littermate controls or commercially bred wild-type C57BL/6J mice (Fig. 8a). While cell lysis was ninjurin-1-dependent, TO-PRO-3 uptake occurred normally in the absence of ninjurin-1, indicating the formation of gasdermin D pores is ninjurin-1-independent (Sup Fig. 5a).

Ninjurin-1 is a transmembrane protein that oligomerizes during pyroptosis into large, filamentous assemblies, which are proposed to disrupt the plasma membrane, leading to cell lysis[14].

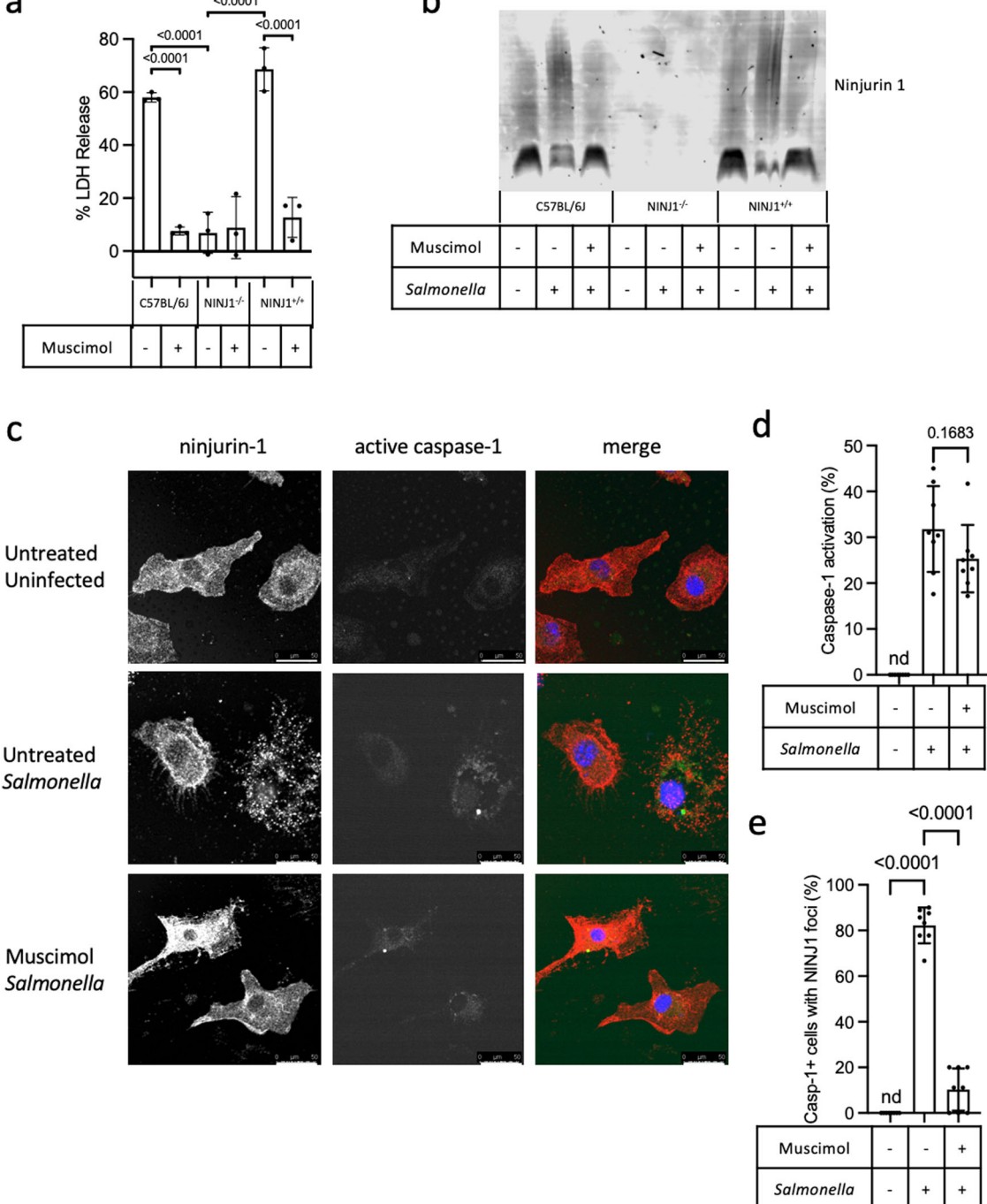

**Fig. 8 Ninjurin-1 oligomerization is blocked by muscimol.** BMM from wild type C57BL/6 J (WT), ninjurin-1$^{-/-}$ (NINJ1$^{-/-}$), and ninjurin$^{+/+}$ (NINJ1$^{+/+}$) littermate controls were infected with *Salmonella* in the presence of muscimol, as indicated. **a** LDH released during cellular lysis was measured at 90 min post infection. **b** BN-PAGE Western blot for ninjurin-1 oligomerization in uninfected, *Salmonella*-infected, and *Salmonella*-infected muscimol-treated cells. Representative blot from two independent experiments. **c** Images representative of 8 taken per condition in 2 independent experiments of NINJ1+/+BMM infected with *Salmonella* for 60 min in the presence or absence of muscimol. Ninjurin-1 oligomerization was visualized as punctate staining with an antibody specific for the N-terminal domain of ninjurin-1 (red). Caspase-1 activation was detected using FAM-YVAD-FMK (green). TO-PRO-3 staining of permeabilized cells (blue) was used to identify nuclei. The percentage of cells containing active caspase-1 was quantified (**d**). The percentage of caspase-1 positive cells containing ninjurin-1 puncta was then quantified (**e**). Statistics: (**a**, **d**, **e**) one-way ANOVA + Tukey's multiple comparisons. ns is not significant. nd is none detected. Data in **a**, **d**, and **e** is from representative experiments (mean ± SD, n = 3 (**a**) or n = 8 (**d** and **e**)) from two independent experiments. Scale bars in **c** represent 50 µm.

Ninjurin-1 oligomerization in pyroptotic cells treated with nigericin or cytoplasmic LPS has been visualized using native electrophoresis-based Western blots[14]. Using this approach, we also observed transition from ninjurin-1 low molecular weight forms to large oligomers in *Salmonella*-infected pyroptotic macrophages (Fig. 8b and Sup Fig. 5b). We found that ninjurin-1 oligomerization in infected cells was inhibited by muscimol (Fig. 8b and Sup Fig. 5b). As a complementary

approach, we used immunofluorescence microscopy, which revealed diffuse ninjurin-1 staining in uninfected control macrophages (Fig. 8c, top row). *Salmonella* infection led to ninjurin-1 redistribution into punctate speck-like assemblies (Fig. 8c, middle row), consistent with oligomers observed by native electrophoresis-based Western blot. While *Salmonella* infection induced equivalent caspase-1 activation in untreated and muscimol-treated cells (Fig. 8c, d), the formation of punctate ninjurin-1 speck-like assemblies was blocked by muscimol (Fig. 8c, e). Together, these results indicate that muscimol inhibits ninjurin-1 oligomerization during pyroptosis to prevent cellular lysis.

The N-terminal extracellular portion of ninjurin-1 has been proposed to form an amphipathic α-helix that inserts into the plasma membrane during pyroptosis and bridges adjacent proteins to form polymers[14,42]. A peptide corresponding to the N-terminal ninjurin-1 amphipathic α-helix is sufficient to directly damage synthetic liposomes and release encapsulated cargo[14]. We observed that ninjurin-1-mediated liposome disruption was not significantly inhibited by muscimol (Sup Fig. 5c), suggesting the possibility that there may not be a direct interaction between muscimol and this portion of ninjurin-1. Taken together, the results presented here indicate that muscimol inhibits ninjurin-1 oligomerization to prevent cellular lysis during pyroptosis, potentially via an indirect mechanism.

## Discussion

Pore formation during pyroptosis was previously proposed to cause plasma membrane rupture solely via osmotic forces, but the discovery that ninjurin-1 is required for plasma membrane rupture downstream of gasdermin D pores opened the possibility that targeting ninjurin-1 oligomerization could prevent lysis and release of pathogenic cellular contents. In this study, we observed that muscimol prevents cellular lysis via a reversible mechanism that is mediated by preventing ninjurin-1 oligomerization. Structure-activity relationship studies revealed that modification of the isoxazole portion of muscimol does not alter the ability of muscimol to prevent cellular lysis, while modification of the side groups of muscimol abolishes protection. Finally, muscimol acts in vivo to reduce LPS-induced lethality and organ dysfunction in a mouse model of septic shock, suggesting the possibility that this approach could provide therapeutic benefit in the many prevalent human diseases that involve pyroptosis.

Muscimol is well-studied as an exogenous agonist for neuronal $GABA_A$ receptors, for which GABA is the endogenous ligand. We previously found that other $GABA_A$ receptor agonists, including GABA itself, and the conformationally restrained muscimol analog, 4,5,6,7-tetrahydroisoxazolo(5,4-c)pyridin-3-ol (THIP) have no activity to prevent pyroptotic lysis[24]. Here, we tested a comprehensive panel of additional muscimol analogs and found additional $GABA_A$ receptor agonists that fail to protect against plasma membrane rupture. We also observed that the protective effect of muscimol is not antagonized by either gabazine or picrotoxin, together supporting the hypothesis that targeting $GABA_A$ receptors is not the mechanism of inhibition of plasma membrane rupture. Instead, we found that muscimol prevents ninjurin-1 oligomerization, as assessed by both native electrophoresis and immunofluorescence. Although the results from our liposome leakage experiments do not support a direct interaction between muscimol and the extracellular α-helical portion of ninjurin-1, we cannot exclude direct interaction with the native full-length protein in the context of the plasma membrane. The processes that trigger ninjurin-1 oligomerization during pyroptosis remain unclear and muscimol could instead indirectly block ninjurin-1 oligomerization by affecting these processes. For example, alterations in membrane curvature or lipid composition can be recognized by amphipathic helices[43] and may initiate a conformational change in ninjurin-1, causing the extracellular α-helical portion to insert into the plasma membrane. Further study will be necessary to better understand these mechanisms.

Early studies of pyroptosis demonstrated that the simple amino acid, glycine, also prevents pyroptotic plasma membrane rupture without affecting antecedent pore formation or IL-1β release[13]. Similar to our findings with muscimol, glycine protection is reversible, and pyroptotic cells protected from lysis by glycine are intact, but metabolically inactive and can be considered dead[27]. Glycine administration is remarkably protective against organ damage and lethality in models of polymicrobial sepsis[44] and LPS endotoxemia[45,46] similar to our findings with muscimol. Recent findings also demonstrate that like muscimol, glycine prevents ninjurin-1 oligomerization in pyroptotic cells, without blocking ninjurin-1 peptide-mediated liposome disruption[47]. To better understand how glycine prevents plasma membrane rupture, we previously tested related small molecules for this effect and identified specific structural determinants, including a requirement for the primary amine and carboxyl groups[24]. Curiously, muscimol has some structural similarity to glycine (Sup Fig. 5D), with a primary amine and a 3-hydroxyisoxazole as a carboxylic bioisostere, raising the possibility of a common molecular interaction with a shared, as of yet, unidentified target. If future studies identify target protein(s), the structure-activity relationships defined here for muscimol and previously for glycine[24] could enable molecular modeling of potential binding sites that conform to the strict molecular constraints demonstrated for inhibition of pyroptotic lysis.

Given the importance of inflammasome and caspase-1 activation in human disease, strategies to inhibit these processes have been actively studied[21,22]. However, blocking inflammasomes or caspase-1 would also limit IL-1β and IL-18, which play essential roles in host defense. Therefore, specifically preventing membrane rupture could be an approach to limit the harmful consequences of pyroptosis, without reducing protective cytokines and pathogen trapping. In addition, it is increasingly well-appreciated that cell death pathways are redundant and interconnected. For example, caspase-1 inactivation does not prevent inflammasome triggered cell death but redirects cells to an alternative pathway of lytic death[23]. Therefore, preventing the common final step of membrane rupture by targeting ninjurin-1 could limit the harmful consequences of lytic cell death to propagate inflammation, without preventing appropriate degradation of the damaged cell corpse and destruction of trapped pathogens. In support of this, ninjurin-1-deficiency and an anti-ninjurin-1 antibody that blocks oligomerization protect against inflammation and tissue injury, although mortality is not delayed, in extreme mouse models of liver damage[48]. Ninjurin-1-deficient mice were not protected from mortality associated with a dose of LPS (54 mg/kg)[14] that was higher than the 10 mg/kg we administered in our experiments (Fig. 3). The discrepancy regarding mortality may reflect differences between these model systems, but we can't presently exclude that the protective effect we observed with muscimol may not be mediated through ninjurin-1 alone. Although we cannot formally exclude a role for GABA receptors in this effect, experimentally increasing GABA concentrations enhances susceptibility to endotoxic lethality[49], suggesting that muscimol's GABA agonist activity is unlikely to protect against LPS-induced organ damage and lethality. Neither muscimol nor glycine themselves likely represent viable therapeutic candidates, due to the relatively high concentrations required and other biologic activities. However, further study of their mechanism may reveal insights into the process of pyroptotic lysis and identify novel small molecules with specific and

potent activity to prevent plasma membrane rupture, which could provide novel therapies for pyroptosis-mediated pathology.

## Methods

**Mouse strains**. C57BL/6J (stock no: 000664, RRID:IMSR_JAX:000664), C57BL/6J ASC-Citrine (stock no: 030744, RRID:IMSR_JAX:030744), Gasdermin D$^{-/-}$ (stock no: 32410, RRID:IMSR_JAX:032410), and BALB/c (stock no: 000651, RRID:IMSR_JAX:000651) mice were purchased from Jackson Laboratories. Caspase-1/11$^{-/-}$ mice were a kind gift from Dr. Richard A. Flavell, Yale University, and bred on-site. Ninjurin-1$^{+/-}$ mice were a kind gift from Genentech and bred on-site to generate Ninjurin-1$^{-/-}$ and Ninjurin-1$^{+/+}$ litter mate controls. Both male and female mice 8–10 weeks of age were used in this study. Mice were housed in specific pathogen-free conditions according to the University of Washington Institutional Animal Care and Use Committee guidelines. We have complied with all relevant ethical regulations for animal testing.

**Cell culture**. Bone marrow-derived macrophages (BMM) were isolated from femur exudates and differentiated at 37 °C in 5% $CO_2$ in Dulbecco's modified Eagle medium supplemented with 5 mM HEPES, 0.05 mM 2-mercaptoethanol, 100 U/mL penicillin/streptomycin, 10% Serum Plus II (Sigma) and containing 30% L929 conditioned medium for 7 days with additional medium added on day 3 or 4. L929 conditioned medium was generated using L929 cells (ATCC Cat no CCL-1, RRID:CVCL_0462) grown to confluency and incubated for 2 additional days before collecting the media. Differentiated BMM were collected by washing with ice-cold PBS containing 1 mM EDTA and resuspended in 10 mL supplemented antibiotic-free medium (without phenol red) containing 5% Serum Plus II and counted. Cells were diluted to $4*10^5$ cells/mL and seeded using the following volumes: 100 µL/well: 96-well plate. 1 mL/well: 24-well plate. For 6-well plates and 6 cm petri dishes, $3–5*10^6$ cells were seeded per well.

**Inflammasome activation**. Late-log cultures of *Salmonella enterica* serovar Typhimurium strain SL1344 (a kind gift from Dr. Brad T. Cookson, University of Washington) were grown in L-broth containing 0.3 M NaCl with shaking at 37 °C. Bacteria were washed, resuspended in PBS, and added at a multiplicity of infection of 10 bacteria per macrophage. For activation of NLRP1b, BMM from BALB/c mice were exposed to 1 µg/mL anthrax lethal toxin, prepared as a 1:1 mix of protective antigen and lethal factor (List Biological). For NLRP3 activation, BMM from C57BL/6 was primed with 100 ng/mL LPS followed by 10 µM Nigericin. For post-apoptotic lysis, BMM were exposed to 1 µM staurosporine for 9 h. Glycine (5 mM), muscimol (1 mM), or muscimol analogs (indicated concentrations) were added at the time of treatment. Picrotoxin (1 mM)[50], GABA (100 mM)[24], or gabazine (10 µM)[51] were added at the same time as muscimol. Biotin-muscimol was obtained from Santa Cruz Biotechnologies and compounds used in Figs. 4, 5, 6, and Sup Fig. 3 were synthesized as previously described in the literature.

**IL-1β enzyme-linked immunosorbent assay (ELISA)**. The level of secreted IL-1β was determined by ELISA using purified anti-mouse IL-1β antibody (Biolegend) as the capture antibody and polyclonal IL-1β antibody conjugated to biotin (ThermoFisher) as the detection antibody. Streptavidin-HRP (Biolegend) and TMB solution (ThermoFisher) were used as detection reagents.

**LDH release assay**. Release of cytoplasmic LDH was measured using the Cytotox-96 non-radioactive kit (Promega). Samples were processed according to the manufacturer's protocol. For each condition, 9 wells were seeded (3 spontaneous lysis, 3 total lysis, and 3 experimental). $OD_{490}$ was measured using a Spectramax M3 plate reader. Cytotoxicity was calculated as ((ODexperimental – ODspontaneous)/(ODmaximum – ODspontaneous)) x 100.

**Immunofluorescence microscopy**. For antibody labeling, BMM were seeded on coverslips in 24-well plates. After caspase-1 activation, cells were fixed and permeabilized with CytoFix/CytoPerm (BD). Caspase-1 activation was detected using FAM-YVAK-FMK (Immunochemistry Technologies). Following fixation, cells were probed with primary anti-mouse ninjurin-1 monoclonal antibody (1:1000; a kind gift from Genentech) diluted in Cytowash (BD) + 3% BSA followed by probing with a fluorescently tagged secondary antibody (ThermoFisher). TO-PRO-3 (ThermoFisher) was used as a nuclear stain to determine the total number of cells per image. For kinetic experiments, BMM were seeded in optical 96-well plates. TO-PRO-3 (membrane-impermeable nuclear dye) was added to the culture medium at 1 micromolar to determine the number of cells that developed gasdermin D pores in the plasma membrane. Kinetic microscopy experiments were performed on a Cytation 1 imaging system running Gen5 version 3.11 using a 10x NA 0.3 objective (Biotek/Agilent). Data collected on the Cytation 1 were quantified using the Gen5 software. Imaging of ninjurin-1 oligomerization was performed on a Leica SP8X confocal system running the LasX imaging software using a 63x NA 1.4 oil immersion objective. Ninjurin-1 oligomerization and identification of FAM-YVAD-FMK positive cells were derived from counting eight high-powered fields per experiment. Total cell counts were used to calculate the percentage of FAM-YVAD-FMK positive cells and the percentage of FAM-YVAD-FMK positive cells with ninjurin-1 foci.

**Expression of GABA receptor isoforms**. RNA was isolated from the brain and BMM from C57BL/6 and BALB/c mice. cDNA was generated using the iScript cDNA synthesis kit (Bio-Rad) and a qPCR reaction was performed using SYBR-green mastermix (Bio-Rad) and primers for each GABA-receptor isoform (see table below). Following the qPCR samples were separated on a 3% agarose gel.

**Primers**. Primers are from a prior study[52] and are provided in Table 1.

**LPS-induced sepsis in mice**. C57BL/6J mice (8–10 weeks old) were treated with 3 mg/kg muscimol in PBS or vehicle by subcutaneous injection. Sepsis was induced by intraperitoneal injection of 10 mg/kg Ultrapure LPS (*Escherichia coli* O55:B5). Mice were weighed daily and euthanized when weight loss was 20% or more of the starting body weight. In addition, mice were scored for appearance, activity, level of consciousness, response to stimulus, eye health, respiratory rate, and respiratory quality. Each category was given a 0–4 score. If a mouse reached a combined score of 12 or higher it was euthanized independent of body weight. These two sets of criteria were used to determine the survival rate[53]. Blood samples were collected at the time of euthanasia or at 96 h post LPS challenge and allowed to clot at room temperature. Sera obtained after centrifugation were analyzed for blood urea nitrogen at Phoenix Central Laboratories (Mukilteo, WA).

**Western blot**. For SDS-PAGE, BMM was lysed with 2x SDS-PAGE Sample buffer (Bio-Rad). Proteins were separated using Any-Kd gels (Bio-Rad), transferred to polyvinylidene fluoride

**Table 1 Primers used in this study.**

| Gene | Forward | Reverse |
|------|---------|---------|
| GABAA a1 | TGCTGGACGGTTATGACAAT | GAAACTGGTCCGAAACTGGT |
| GABAA a2 | ACAACCTTGAGCATCAGTGC | AATTCACGGTTGCAAATTCA |
| GABAA a3 | GACAGTCCTGCTGAGACCAA | ATAGCTGATTCCCGGTTCAC |
| GABAA a4 | AGAACTCAAAGGACGAGAAATTGT | TTCACTTCTGTAACAGGACCCC |
| GABAA a5 | TCCATTGCACACAACATGAC | GCAGAGATTGTCAGACGCAT |
| GABAA a6 | GGTGACCGGGCATCCCAGTGA | TGTTACAGCACCCCCAAATCCTGGC |
| GABAA b1 | GGTTTGTTGTGCACACAGCTCC | ATGCTGGCGACATCGATCCGC |
| GABAA b2 | AGCTGCTAATGCCAACAATG | GTCCCATTACTGCTTCGGAT |
| GABAA b3 | CAAAGCCATCGACATGTACC | CTTCTCCGCAAGCTTCTTCT |
| GABAA g1 | ATCCACTCTCATTCCCATGAACAGC | ACAGAAAAAGCTAGTACAGTCTTTGC |
| GABAA g2 | TGGTCACCGAATGTGTTTCT | TACTTTGCCTTGCAGGTTTG |
| GABAA d | TCAAATCGGCTGGCCAGTTCCC | GCACGGCTGCCTGGCTAATCC |
| GABAA e | ACTGCGCCCTGGCATTGGAG | AGGCCCGAGGCTGTTGACAA |
| GABAA q | GCTGGAGGTGGAGAGCTATGGCT | CCCCAGGTACGTGTACTGAGGGA |
| GABAA p | TCGGTGGTGACCCAGTTCGGAT | TCTGTCCAACGCTGCCGGAG |
| GABAA r1 | CCATCTAGGAAAGGCAGCAG | GAGCTTCGTCTCAGGATTGG |
| GABAA r2 | GCTGCCTGTTGCATCATAGA | ATACAAATGGCTTGGCTTGG |
| GABAA r3 | CAACTCAACAGGAGGGGAAA | TCCACATCAGTCTCGCTGTC |
| GABAB1 | ACGTCACCTCGGAAGGTTG | CACAGGCAGGAAATTGATGGC |
| GABAB2 | CAGCAAGCGTTCGGGTGTA | GTCTTGGCGATGACCCAGAT |
| Beta-actin | CACTGTCGAGTCGCGTCC | TCATCCATGGCGAACTGGTG |

(PVDF), and analyzed with anti-HMGB1 (1:1000; Abcam, ab79823, RRID: AB_1603373) or anti-GSDMD (1:1000; Cell Signaling Technologies, 39754, RRID: AB_2916333) antibodies. Secondary antibodies were purchased from LiCor and used at 1:10,000. Blots were scanned on a Licor Odyssey and Image Studio Light was used to quantify band intensity. For blue native PAGE, BMM was lysed using 1% digitonin, 150 mM NaCl, 50 mM Tris. 4x lysis buffer (ThermoFisher) and G-250 were added to the samples prior to loading on a 3–12% Native PAGE gel. After separation, proteins were transferred to PVDF and analyzed using anti-ninjurin-1 polyclonal antibody (1–2 μg/mL; a kind gift from Genentech).

**Liposomal cargo release assay.** 1,2-dioleoyl-*sn*-glycero-3-phos-phocholin and 1,2-dioleoyl-*sn*-glycero-3-phospho-L-serine (DOPC:-DOPS), 80:20) loaded with 5(6)-carboxy fluorescein were purchased from Encapsula Nano Sciences. Washed liposomes were exposed to 0.5 mg/mL ninjurin-1 peptide (HYASKKSAAESMLDIALLMA-NASQLKAVVE; GenScript Biotech Corporation) at a 1:1 ratio in the absence or presence of 1 mM muscimol. After 1 h of incubation, the fluorescent intensity was measured. Spontaneous and total lysis (Triton X-100) were used to calculate the percentage of cargo release for each condition.

**Cell Titer Glo assay.** ASC-Citrine C57BL/6 BMM were infected for 2 h with *Salmonella* in the presence or absence of 1 mM muscimol. At 2 h an equal volume of Cell Titer Glo 2.0 reagent (Promega, Cat no G9242) was added. After 10 min luminescence was determined using a SpectraMax M3 plate reader. Viability was calculated as a percentage of untreated uninfected cells.

**Statistics and reproducibility.** Statistical analysis was performed in Prism (GraphPad Software, v9.3). The statistical tests used are indicated in each figure legend. Each data point is from a distinct sample and no repeated measurements were taken. All data are presented as mean +/− SD.

**Reporting summary.** Further information on research design is available in the Nature Portfolio Reporting Summary linked to this article.

**Data availability**

The source data presented in this manuscript are provided as Suppl Data 1. The uncropped blot for Fig. 8b is provided in Fig S5b. Gasdermin D (Fig. 1d) and HMGB1 (Fig. 2b) Westerns are shown as uncropped/unedited images obtained directly from the instrument.

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

## Acknowledgements

S.L.F. was supported by the National Institute of Allergy and Infectious Diseases of the National Institutes of Health under Award Numbers R01AI162684, R21AI153487, and R21AI178367. The content is solely the responsibility of the authors and does not necessarily represent the official views of the National Institutes of Health. We thank Dr. Richard A. Flavell (Yale University) for caspase-1/11$^{-/-}$ mice, Genentech for ninjurin-1$^{+/-}$ mice and anti-ninjurin-1 antibody, and Dr. Brad T. Cookson (University of Washington) for *Salmonella* strain SL1344. Microscopy using the Leica SP8X was performed at the W.M. Keck Microscopy Center with the support of NIH award S10OD016240.

## Author contributions

A.B.D., W.P.L., M.J.A., B.F., and S.L.F. conceived of the study, designed experiments, and interpreted results. A.B.D., W.P.L., and M.J.A. performed experiments. B.F. provided muscimol analogs. A.B.D. and S.L.F. wrote an initial draft and worked with W.P.L., M.J.A., and B.F. to complete the manuscript.

## Competing interests

The authors declare no competing interests.
