## [Peer Review File · Communications Biology]

Reviewers' comments:

Reviewer #1 (Remarks to the Author):

In this study, the authors follow up on previous work showing that the small molecule muscimol blocks pyroptotic cell death. They show that muscimol does not affect ASC specks formation and IL-1b release, but blocks IL-1a and HMGB1 release, thus acts downstream of GSDMD pore formation. They show that muscimol blocks LPS induced lethality in vivo, and that the protection against cell lysis is reversible. As muscimol is known to be a GABA_A receptor antagonist, they investigate the function of these in pyroptosis. Muscimol protection is not altered by GABA_A receptor antagonists or recapitulated by

9 other GABA_A agonists, suggesting that muscimol acts via a novel mechanism. Screening muscimol-related compounds shows that distinct molecular determinants define inhibition of pyroptosis and GABA_A receptors. Finally, the authors show that muscimol prevents NINJ1 clustering upon pyroptosis induction, suggesting that it acts similarly as glycine treatment.

The study is well written and the results are convincing. The data show that muscimol somehow affect NINJ1 oligomerization and that this is the most likely explanation for its pyroptosis-blocking effect. Whether this inhibition is direct or indirect remains unclear though. Some additional controls would be necessary though (see below) to complete the story.

My largest point of criticism is to what degree the results will be useful or interesting to the field. Glycine has been used by many researchers to block pyroptosis, and knowing that it acts at the level of NINJ1 has a certain value, even though it doesn't explain neither how NINJ1 is activated nor how this is affected by glycine. Muscimol on the other hand is not commonly used to block pyroptosis and thus the manuscript at hand will only be of limited interest to the field.

Comments:

Fig. 1: It would be nice to expand beyond the NLRC4 inflammasome to other inflammasomes, and also apoptosis or pore-driven cell death.

The authors should also include GSDMD cleavage as another readout of pore formation to show that all step upstream of GSDMD pore formation remain unaffected.

The blots in Fig. 2B are rather poor quality. I would suggest to repeat these blots.

The authors show that Muscimol blocks LPS lethality in vivo; and relate this to the effect of Muscimol on NINJ1. This is however in contrast to data published by Kayagaki et al. 2021, that reports that NINJ1 deficiency in mice has no effect on LPS lethality. It is thus possible that Muscimol acts unspecifically on other pathways in vivo and not on NINJ1 alone.

It would be good if the imaging analysis would be complemented by other data. Ideally a biochemical analysis, such as NIN1 oligomerization on native PAGE, as other studies have shown.

Reviewer #2 (Remarks to the Author):

Building on their own work (Loomis et al Cell Death Dis 2019) and that of others (Kayagaki et al Nature 2021 and Borges et al eLife 2022), in the submitted manuscript, the authors observed that muscimol inhibits pyroptosis-induced lytic cell death downstream of gasdermin by interfering with NINJ1-mediated membrane rupture. This primary finding is complemented by structure-function studies directed at identifying the molecular features of muscimol that bestow its membrane rupture

inhibitory function.

Overall, the topic of lytic cell death and the mechanism of NINJ1 are of broad interest and timely. To that end, the manuscript provides additional insight on the pharmacologic targeting of NINJ1 and positions muscimol as a potential structural backbone on which cytoprotectant agents can be designed. The structure-function studies are particularly commendable. In its current form, however, the manuscript falls short of directly linking the membrane-protective effect of muscimol with NINJ1.

Major points:

1. Whether muscimol directly or indirectly targets NINJ1 was not examined. Determining whether the effect of muscimol is direct or indirect would be hugely impactful to the field of lytic cell death at large. A mechanistic understanding provided on how muscimol inhibits NINJ1-mediated plasma membrane rupture should be explored. For example, biophysical methods could be useful in defining a direct interaction between NINJ1 and muscimol. Can direct binding of muscimol (and its structural derivatives) be measured through bilayer interferometry or surface plasmon resonance? If so, what are the key NINJ1 residues / domains required for binding? These could provide invaluable insight into NINJ1 clustering, whose mechanism remains unclear. On the other hand, if muscimol does not bind directly, exploration of this indirect mechanism should be provided.

2. Muscimol's inhibitory effect on NINJ1 oligomerization involves fluorescence microscopy-based studies (Fig 8). These studies are somewhat limited but form the basis for the claims that muscimol is interfering with NINJ1 oligomerization. Additional information is needed to support this claim. Do the authors know whether these puncta represent true oligomers or NINJ1-containing complexes? Is this clustering within the plasma membrane or delivery from endomembrane compartments? If the latter, could muscimol be interfering with NINJ1 delivery to the plasma membrane? More information related to the quantification (definition of NINJ1 puncta and how they were evaluated) would be beneficial. Moreover, to complement the microscopy-based approach, I would suggest also investigating NINJ1 clustering by blue native-PAGE as was done in both the Kayagaki et al (Nature 2021) and Borges et al (eLife 2022) studies.

3. The discussion around the authors' observations of muscimol and the known effect of glycine (lines 226 and 235) are fascinating and worthy of expansion – with data and/or an expanded text. Short of the biophysical studies suggested above, is there more insight that can be gleaned from the reported structure-function studies in the submitted manuscript, glycine's cytoprotective effect, and predicted NINJ1 structure that would provide either (1) the molecular framework for NINJ1 inhibitors; and (2) the mechanism of NINJ1 clustering? For the former, can it be used to predict an alternative pharmacologic NINJ1 inhibitor that can be tested empirically?

4. Statistical descriptions and analyses are not always made clear:

- Number of independent experiments are inconsistently listed. They should be indicated throughout in the Figure legends.

- Consider depicting the individual data points (as done in Fig 3B). This will greatly improve the readability of the data.

- In some experiments (e.g. Fig 8 and Suppl Fig 2C), it appears that $N = 2$ independent experiments were conducted for some groups. In those cases, is ANOVA an appropriate statistical test? If minimum sample size criteria cannot be met, consider additional independent experiments or an alternative statistical test that can accommodate $N = 2$.

Minor points:

1. Presumably, the cells induced to undergo pyroptosis but treated with muscimol are no longer viable. Can the authors provide evidence that the cells treated with muscimol have lost viability but have maintained plasma membrane integrity?

2. Consider updating the reference list to include the recent study by Borges et al (eLife 2022; PMID 3646882). For example, it would be an appropriate paper for discussion / reference on line 92 and 196 among other places.

3. Scale bar missing in Fig 8C.

4. Supplemental Fig 2: It appears like the same data was used for the Salmonella-treated conditions with and without muscimol (columns #4 and #5). This should be clarified and corrected if that was indeed the case. Showing the individual data points from individual experiments, as suggested above, would help mitigate against that type of confusion. Also, while it won't have huge bearing on the interpretation of these data, is there a reason why GABA wasn't used alone in the salmonella-treated cells?

5. I did not see a listing of the NINJ1 antibody that was used other than it was supplied by Genentech, Inc. In the Kayagaki et al (Nature 2021) paper, they describe both a polyclonal and monoclonal antibody. Have the authors validated the antibody (e.g. in their KO animals)?

6. It would be helpful if the authors provide the dilutions / concentrations of the antibodies used in their studies.

Response to Reviewers for COMMSBIO-23-0024-T

Muscimol Inhibits Plasma Membrane Rupture and Ninjurin-1 Oligomerization During Pyroptosis

Dear Dr. Fink,

Your manuscript entitled "Muscimol Inhibits Plasma Membrane Rupture and Ninjurin-1 Oligomerization During Pyroptosis" has now been seen by 2 referees, whose comments are appended below. You will see from their comments copied below that while they find your work of potential interest, they have raised quite substantial concerns that must be addressed. In light of these comments, we cannot accept the manuscript for publication, but would be interested in considering a revised version that addresses these serious concerns.

We hope you will find the referees' comments useful as you decide how to proceed. Should further experimental data or analysis allow you to address these criticisms, we would be happy to look at a substantially revised manuscript. However, please bear in mind that we will be reluctant to approach the referees again in the absence of major revisions.

In particular, please note that the following revisions would be necessary for us to contact our referees again: (1) GSDMD cleavage as a readout for pore formation, and (2) NINJ1 clustering by blue native-PAGE. (3) Additional investigation of an interaction between NINJ1 and muscimol (e.g., bilayer interferometry, SPR) would be highly appreciated.

We are committed to providing a fair and constructive peer-review process. Do not hesitate to contact us if you wish to discuss the revision or if there are specific requests from the reviewers that you believe are technically impossible or unlikely to yield a meaningful outcome.

We thank the editor and reviewers for their positive comments and thoughtful suggestions to improve our manuscript. As detailed further below, our revised manuscript now addresses all of these comments and includes new results, which support and extend our previous conclusions. In particular, we provide 1) GSDMD cleavage in revised Figure 1D, 2) NINJ1 clustering by blue native-PAGE in revised Figure 8B, and 3) additional investigation of an interaction between NINJ1 and muscimol using ninjurin-1-mediated liposome leakage assays (revised Supplemental Figure 5C).

Reviewer #1 (Remarks to the Author):

In this study, the authors follow up on previous work showing that the small molecule muscimol blocks pyroptotic cell death. They show that muscimol does not affect ASC specks formation and IL-1b release, but blocks IL-1a and HMGB1 release, thus acts downstream of GSDMD pore formation. They show that muscimol blocks LPS induced lethality in vivo, and that the protection against cell lysis is reversible. As muscimol is known to be a GABA_A receptor antagonist, they investigate the function of these in pyroptosis. Muscimol protection is not altered by GABA_A receptor antagonists or recapitulated by 9 other GABA_A agonists, suggesting that muscimol acts via a novel mechanism. Screening muscimol-related compounds shows that distinct molecular determinants define inhibition of pyroptosis and GABA_A receptors. Finally, the authors show that muscimol prevents NINJ1 clustering upon pyroptosis induction, suggesting that it acts similarly as glycine treatment.

The study is well written and the results are convincing. The data show that muscimol somehow affect NINJ1 oligomerization and that this is the most likely explanation for its pyroptosis-blocking effect. Whether this inhibition is direct or indirect remains unclear though. Some additional controls would be necessary though (see below) to complete the story. My largest point of criticism is to what degree the results will be useful or interesting to the field. Glycine has been used by many researchers to block pyroptosis, and knowing that it acts at the level of Ninjurin-1 has a certain value, even though it doesn't explain neither how NINJ1 is activated nor how this is affected by glycine. Muscimol on the other hand is not commonly used to block pyroptosis and thus the manuscript at hand will only be of limited interest to the field.

We thank the reviewer for their positive comments and suggestions of additional controls to complete the story, which we have added as described below. We agree that muscimol is not as commonly used as an experimental tool to block pyroptotic lysis as glycine, but propose that the increased potency and molecular complexity compared to glycine could facilitate further mechanistic insights into inhibition of

ninjurin-1 oligomerization, and as noted by Reviewer #2, this manuscript positions muscimol as a potential structural backbone on which cytoprotectant agents can be designed. Thus, we propose that the novel data presented here will be of interest to advance the field as appropriate for *Communications Biology*.

Comments:

Fig. 1: It would be nice to expand beyond the NLRC4 inflammasome to other inflammasomes, and also apoptosis or pore-driven cell death.

As suggested, we included experiments with NLRP1b inflammasome activation using anthrax lethal toxin (Figure 1A and Figure 7C), NLRP3 inflammasome activation using nigericin in LPS primed macrophages (Figure 1A) and apoptosis induction with staurosporine (Figure 1A). Consistent with the requirement for ninjurin-1 for plasma membrane rupture stimulated by multiple inflammasomes and in apoptotic cells (PMID: 33472215), we find that muscimol inhibits plasma membrane rupture across pyroptotic stimuli and in apoptotic cells.

The authors should also include GSDMD cleavage as another readout of pore formation to show that all steps upstream of GSDMD pore formation remain unaffected.

As suggested, we added Western blot-based assessment of GSDMD cleavage (Figure 1D) as another readout of pore formation, which further supported our previous results that steps upstream of GSDMD pore formation are not affected by muscimol.

The blots in Fig. 2B are rather poor quality. I would suggest to repeat these blots.

As suggested, we repeated this experiment and include higher quality blots in Figure 2B.

The authors show that Muscimol blocks LPS lethality *in vivo*; and relate this to the effect of Muscimol on Ninjurin-1. This is however in contrast to data published by Kayagaki et al. 2021, that reports that Ninjurin-1 deficiency in mice has no effect on LPS lethality. It is thus possible that Muscimol acts unspecifically on other pathways *in vivo* and not on NINJ1 alone.

We agree that it is quite possible that muscimol has activities *in vivo* in addition to blocking ninjurin-1 oligomerization and appreciate the opportunity to discuss this further.

Line 284: Therefore, preventing the common final step of membrane rupture by targeting ninjurin-1 could limit the harmful consequences of lytic cell death to propagate inflammation, without preventing appropriate degradation of the damaged cell corpse and destruction of trapped pathogens. In support of this, ninjurin-1-deficiency and an anti-ninjurin-1 antibody that blocks oligomerization protect against inflammation and tissue injury, although mortality is not delayed, in extreme mouse models of liver damage⁴⁸. Ninjurin-1-deficient mice were not protected from mortality associated with a dose of LPS (54 mg/kg)¹⁴ that was higher than the 10 mg/kg we administered in our experiments (Fig 3). The discrepancy regarding mortality may reflect differences between these model systems, but we can't presently exclude that the protective effect we observed with muscimol may not be mediated through ninjurin-1 alone. Although we cannot formally exclude a role for GABA receptors in this effect, experimentally increasing GABA concentrations enhances susceptibility to endotoxic lethality⁴⁹, suggesting that muscimol's GABA agonist activity is unlikely to protect against LPS-induced organ damage and lethality. Neither muscimol nor glycine themselves likely represent viable therapeutic candidates, due to the relatively high concentrations required and other biologic activities.

It would be good if the imaging analysis would be complemented by other data. Ideally a biochemical analysis, such as NIN1 oligomerization on native PAGE, as other studies have shown.

We appreciate the opportunity to use an orthogonal approach to assess NIN1 oligomerization. As suggested, we added native electrophoresis-based Western blots (Figure 8B, Supp Fig 5B), which demonstrated the transition from ninjurin-1 low molecular weight forms to large oligomers during pyroptosis and provide additional support for our previous conclusion, based on immunofluorescence, that muscimol inhibits ninjurin-1 oligomerization.

Reviewer #2 (Remarks to the Author)

Building on their own work (Loomis et al Cell Death Dis 2019) and that of others (Kayagaki et al Nature 2021 and Borges et al eLife 2022), in the submitted manuscript, the authors observed that muscimol inhibits pyroptosis-induced lytic cell death downstream of gasdermin by interfering with NINJ1-mediated membrane rupture. This primary finding is complemented by structure-function studies directed at identifying the molecular features of muscimol that bestow its membrane rupture inhibitory function.

Overall, the topic of lytic cell death and the mechanism of NINJ1 are of broad interest and timely. To that end, the manuscript provides additional insight on the pharmacologic targeting of NINJ1 and positions muscimol as a potential structural backbone on which cytoprotectant agents can be designed. The structure-function studies are particularly commendable. In its current form, however, the manuscript falls short of directly linking the membrane-protective effect of muscimol with NINJ1.

We thank the reviewer for their positive comments and thoughtful suggestions to improve our manuscript.

Major points:

1. Whether muscimol directly or indirectly targets NINJ1 was not examined. Determining whether the effect of muscimol is direct or indirect would be hugely impactful to the field of lytic cell death at large. A mechanistic understanding provided on how muscimol inhibits NINJ1-mediated plasma membrane rupture should be explored. For example, biophysical methods could be useful in defining a direct interaction between NINJ1 and muscimol. Can direct binding of muscimol (and its structural derivatives) be measured through bilayer interferometry or surface plasmon resonance? If so, what are the key NINJ1 residues / domains required for binding? These could provide invaluable insight into NINJ1 clustering, whose mechanism remains unclear. On the other hand, if muscimol does not bind directly, exploration of this indirect mechanism should be provided.

We appreciate this comment and agree that these are critical questions that we are working to answer. First, we used circular dichroism to ask whether muscimol alters the alpha helical structure of the N-terminal region required for lysis and observed the characteristic absorption pattern for α -helices with two minima at 208 nm and 222 nm (see Figure). Muscimol did not alter the pattern of these dips, suggesting that it does not alter the α -helical secondary structure, although the inherent ability of muscimol to absorb ultraviolet light (PMID: 2857769) interfered with circular dichroism measurements at lower wavelengths, thus limiting our ability to draw robust conclusions and perhaps not suitable for publication.

Circular dichroism spectra of ninjurin-1 peptide in the presence of muscimol or vehicle control, as indicated. Measurements obtained when the corresponding HT voltage was above the instrument manufacturer's recommended cutoff of 600V, up to a maximum of 1000V, are indicated by lighter colors.

Then, we tested whether muscimol blocks liposome rupture and release of encapsulated cargo induced directly by a peptide corresponding to the N-terminal ninjurin-1 amphipathic α -helix. We found that ninjurin-1-mediated liposome disruption was not significantly inhibited by muscimol (Supplemental Figure 5C). Together, these results suggest that there may not be a direct interaction between muscimol and this portion of ninjurin-1, although we acknowledge that there are limitations to these experiments. Line 252: "Although the results from our liposome leakage experiments do not support a direct interaction between muscimol and the extracellular α -helical portion of ninjurin-1, we cannot exclude direct interaction with the native full-length protein in the context of the plasma membrane. The processes that trigger ninjurin-1 oligomerization during pyroptosis remain unclear and muscimol could instead indirectly block ninjurin-1 oligomerization by affecting these processes. For example, alterations in membrane curvature or lipid composition can be recognized by amphipathic helices⁴³ and may initiate a conformational change in ninjurin-1, causing the extracellular α -helical portion to insert into the plasma membrane. Further study will be necessary to better understand these mechanisms."

To further study this, we also engaged an expert collaborator with specialized experience using biophysical approaches to identify and characterize biomolecular interactions, including surface plasmon resonance and biolayer interferometry. Unfortunately, the feasibility of both surface plasmon resonance and biolayer interferometry to query direct binding of muscimol with ninjurin-1 is limited. These techniques involve immobilization of probe molecules to a sensor surface; although various coupling strategies can be used, our structure-activity relationship analysis demonstrates that even minor modifications to muscimol's functional groups completely abolish activity against pyroptosis (Figures 5, 6 and 7, and Supplemental Figure 4). Thus, an approach to immobilize muscimol without disrupting target interaction was considered technically not feasible. As an alternative, we considered immobilizing ninjurin-1 and/or ninjurin-1-derived peptides, which could be readily achievable, but would require using muscimol as the soluble ligand molecule. Surface plasmon resonance and biolayer interferometry both fundamentally rely on signal generated by mass addition to detect biomolecular interactions, and the small molecular weight of muscimol means it would be unlikely to generate a robust signal upon binding. Other methods for measuring binding would suffer from the same challenges with either immobilizing or sensing muscimol, since adding detection tags to muscimol would create the same problems as adding a structure to immobilize it.

As yet an alternative approach, we engaged another collaborator with specialized expertise in protein structure modeling to explore the possibility of using published ninjurin-1 structures, including the recently reported cryoEM structure of polymerized ninjurin-1 (PMID: 37198476, PDB Entry - 8CQR) for molecular docking studies with both muscimol and glycine. Given our knowledge of the activity of analogs of muscimol (this study) and glycine (PMID: 30975978), we hoped that any identified potential binding sites could be evaluated for disruption by the molecular changes we have empirically found to disrupt activity, thus providing experimental support for any findings derived from computational modeling. However, the extremely small size of glycine and muscimol, their relatively limited chemical features for molecular interactions and lack of evidence to support putative active site candidates within ninjurin-1 were together considered barriers to gathering robust information or drawing any meaningful conclusions from this approach. We plan to pursue further, more technically specialized and challenging approaches to experimentally identify candidate muscimol and glycine binding sites, but propose that this is beyond the scope of the current study.

2. Muscimol's inhibitory effect on NINJ1 oligomerization involves fluorescence microscopy-based studies (Fig 8). These studies are somewhat limited but form the basis for the claims that muscimol is interfering with NINJ1 oligomerization. Additional information is needed to support this claim. Do the authors know whether these puncta represent true oligomers or NINJ1-containing complexes? Is this clustering within the plasma membrane or delivery from endomembrane compartments? If the latter, could muscimol be interfering with NINJ1 delivery to the plasma membrane? More information related to the quantification (definition of NINJ1 puncta and how they were evaluated) would be beneficial. Moreover, to complement the microscopy-based approach, I would suggest also investigating NINJ1 clustering by blue native-PAGE as was done in both the Kayagaki et al (Nature 2021) and Borges et al (eLife 2022) studies.

We appreciate the opportunity to use an orthogonal approach to assess NIN1 oligomerization. As suggested, we added native electrophoresis-based Western blots (Figure 8B, Supp Fig 5B), which demonstrated the transition from ninjurin-1 low molecular weight forms to large oligomers during pyroptosis and provide additional support for our previous conclusion, based on immunofluorescence, that muscimol inhibits ninjurin-1 oligomerization.

The observation of NINJ1 puncta by immunofluorescence and relationship to oligomerization has been previously proposed by others (PMID: 33472215) and very recently supported by super-resolution microscopy and cryo-EM structure (PMID: 37198476), which we now cite and include in the text. Line 222: "The N-terminal extracellular portion of ninjurin-1 has been proposed to form an amphipathic α -helix that inserts into the plasma membrane during pyroptosis and bridges adjacent proteins to form polymers^{14, 42}." We have also included description of the immunofluorescence microscopy quantification to the Methods.

3. The discussion around the authors' observations of muscimol and the known effect of glycine (lines 226 and 235) are fascinating and worthy of expansion – with data and/or an expanded text. Short of the biophysical studies suggested above, is there more insight that can be gleaned from the reported structure-function studies in the submitted manuscript, glycine's cytoprotective effect, and predicted NINJ1 structure that would provide either (1) the molecular framework for NINJ1 inhibitors; and (2) the mechanism of NINJ1 clustering? For the former, can it be used to predict an alternative pharmacologic NINJ1 inhibitor that can be tested empirically?

We appreciate the reviewer's positive comments and welcome the opportunity to expand on these observations with new text and new Supplemental Figure 5D to better illustrate the similarity between muscimol and glycine.

Beginning at line 262: "Similar to our findings with muscimol, glycine protection is reversible and pyroptotic cells protected from lysis by glycine are intact, but metabolically inactive and can be considered dead²⁷. Glycine administration is remarkably protective against organ damage and lethality in models of polymicrobial sepsis⁴⁴ and LPS endotoxemia^{45, 46} similar to our findings with muscimol. Recent findings also demonstrate that like muscimol, glycine prevents ninjurin-1 oligomerization in pyroptotic cells, without blocking ninjurin-1 peptide-mediated liposome disruption⁴⁷. To better understand how glycine prevents plasma membrane rupture, we previously tested related small molecules for this effect and identified specific structural determinants, including a requirement for the primary amine and carboxyl groups²⁴. Curiously, muscimol has some structural similarity to glycine (Sup Fig 5D), with primary amine and a 3-hydroxyisoxazole as a carboxylic bioisostere, raising the possibility of a common molecular interaction with a shared, as of yet, unidentified target. If future studies identify target protein(s), the structure-activity relationships defined here for muscimol and previously for glycine²⁴ could enable molecular modeling of potential binding sites that conform to the strict molecular constraints demonstrated for inhibition of pyroptotic lysis."

As mentioned previously, we have also explored the possibility of using the ninjurin-1 structure for molecular docking studies with muscimol and glycine. However, we propose that the current utility of this approach is limited, given the reasonable likelihood that muscimol and glycine may indirectly affect ninjurin-1 oligomerization and directly interact with a currently unknown target. In addition, the small size of glycine and muscimol, and somewhat limited chemical features for molecular interactions would limit our ability to derive robust conclusions from modeling efforts, in the absence of a better-defined target. We do, however, propose that the data presented here will enable future computational approaches to understand and model potential binding site interactions, once a target is better defined.

4. Statistical descriptions and analyses are not always made clear:

- Number of independent experiments are inconsistently listed. They should be indicated throughout in the Figure legends.

We have added clear indication of the number of independent experiments in the Figure legends.

- Consider depicting the individual data points (as done in Fig 3B). This will greatly improve the readability of the data.

We appreciate this suggestion and have individual data points to the Figures throughout the manuscript.

- In some experiments (e.g. Fig 8 and Suppl Fig 2C), it appears that N = 2 independent experiments were conducted for some groups. In those cases, is ANOVA an appropriate statistical test? If minimum sample size criteria cannot be met, consider additional independent experiments or an alternative statistical test that can accommodate N = 2.

Thank you for this comment. We have performed additional experiments and revised the Figures to include only data from experiments with N>2.

Minor points:

1. Presumably, the cells induced to undergo pyroptosis but treated with muscimol are no longer viable. Can the authors provide evidence that the cells treated with muscimol have lost viability but have maintained plasma membrane integrity?

As suggested, we assessed viability using the Cell Titer Glo assay, which measures cellular ATP and found reduced cellular ATP content in *Salmonella*-infected cells compared to uninfected controls, which was not rescued by muscimol (Supplemental Figure 1B).

Line 83: "Gasdermin D pores are sufficient to mediate loss of cytosolic ATP and cessation of metabolic activity, indicating cell death, independently of plasma membrane rupture^{27, 28}. We found reduced cellular ATP content in *Salmonella*-infected cells compared to uninfected controls, which was not rescued by muscimol (Sup Fig 1B). These results suggest that muscimol does not affect gasdermin D cleavage, pore formation or consequent cessation of metabolic activity."

2. Consider updating the reference list to include the recent study by Borges et al (eLife 2022; PMID 3646882). For example, it would be an appropriate paper for discussion / reference on line 92 and 196 among other places.

We are glad to include this new study (now reference 47), which was published after our initial submission.

Line 266: "Recent findings also demonstrate that like muscimol, glycine prevents ninjurin-1 oligomerization in pyroptotic cells, without blocking ninjurin-1 peptide-mediated liposome disruption⁴⁷."

3. Scale bar missing in Fig 8C.

We apologize that the scale bars were too small to be appreciated in our first submission and have increased the size in revised Figure 8C.

4. Supplemental Fig 2: It appears like the same data was used for the *Salmonella*-treated conditions with and without muscimol (columns #4 and #5). This should be clarified and corrected if that was indeed the case. Showing the individual data points from individual experiments, as suggested above, would help mitigate against that type of confusion. Also, while it won't have huge bearing on the interpretation of these data, is there a reason why GABA wasn't used alone in the *salmonella*-treated cells?

Yes, this was the case, as the data previously shown as separate graphs in Supplemental Figure 2 were derived from the same large experiment. For clarity, we have combined these data into a single graph, now Supplemental Figure 3, which does not have any repeated data. GABA alone was not used in *Salmonella*-treated cells as our previous study demonstrated that GABA had no effect on LDH release during pyroptosis induced by *Salmonella*, and we have included this in the text.

Line 144: "We previously found that GABA, the physiologic GABAA receptor agonist, had no cytoprotective effect by itself²⁴, and additionally we found that GABA does not interfere with the ability of muscimol to protect against pyroptotic lysis (Supp Fig 3)."

5. I did not see a listing of the NINJ1 antibody that was used other than it was supplied by Genentech, Inc. In the Kayagaki et al (Nature 2021) paper, they describe both a polyclonal and monoclonal antibody. Have the authors validated the antibody (e.g. in their KO animals)?

As suggested, we added specification of the NINJ1 antibodies supplied by Genentech and used in our experiments.

Line 388: "For blue native PAGE, ...proteins were transferred to PVDF, and analyzed using anti-ninjurin-1 polyclonal antibody (1-2 µg/mL; a kind gift from Genentech)." We also included cells from KO animals in the native PAGE Western blot (Figure 8B), which validates specificity of the polyclonal antibody.

Line 346: "Immunofluorescence microscopy. For antibody labeling... cells were probed with primary anti-mouse ninjurin-1 monoclonal antibody (1:1000; a kind gift from Genentech)". Specificity of this monoclonal antibody was validated previously with KO cells (PMID: 33472215).

6. It would be helpful if the authors provide the dilutions / concentrations of the antibodies used in their studies.

We added the dilutions / concentrations of the antibodies in the text.

Line 388: "For blue native PAGE, ...proteins were transferred to PVDF, and analyzed using anti-ninjurin-1 polyclonal antibody (1-2 µg/mL; a kind gift from Genentech)."

Line 346: "Immunofluorescence microscopy. For antibody labeling... cells were probed with primary anti-mouse ninjurin-1 monoclonal antibody (1:1000; a kind gift from Genentech)"

REVIEWERS' COMMENTS:

Reviewer #1 (Remarks to the Author):

The authors present a improved version of their manuscript and have addressed all my previous comments by performing additional experiments. I have no further comments and support to publication of the manuscript in its current form.

Reviewer #2 (Remarks to the Author):

In their revised manuscript, the authors have addressed my initial queries. Their data suggest that muscimol inhibits pyroptosis-induced lytic cell death downstream of gasdermin by interfering with NINJ1-mediated membrane rupture, a finding complemented by structure-function studies. That the mechanism by which muscimol is acting on NINJ1 remains undefined is perhaps not surprising given how little we know about NINJ1 activation and regulation in plasma membrane rupture. This molecular understanding may require further integration with the recent (Degen M et al Nature 2023) and forthcoming structural studies. Nevertheless, by providing further insight on the pharmacologic inhibition of NINJ1, they complement the work of Borges JP et al (eLife 2022) and Kayagaki N et al (Nature 2023) by expanding the armamentarium by which NINJ1 can be targetted in cytoprotective strategies designed. As such, the findings will be of interest to the field of lytic cell death.

One minor correction: Supplemental Fig 3 is missing a label for the y axis.

Response to Reviewers for COMMSBIO-23-0024A

Muscimol Inhibits Plasma Membrane Rupture and Ninjurin-1 Oligomerization During Pyroptosis

REVIEWERS' COMMENTS:

Reviewer #1 (Remarks to the Author):

The authors present a improved version of their manuscript and have addressed all my previous comments by performing additional experiments. I have no further comments and support to publication of the manuscript in its current form.

We thank the reviewer for their positive comments and support to publish our manuscript in its current form.

Reviewer #2 (Remarks to the Author):

In their revised manuscript, the authors have addressed my initial queries. Their data suggest that muscimol inhibits pyroptosis-induced lytic cell death downstream of gasdermin by interfering with NINJ1-mediated membrane rupture, a finding complemented by structure-function studies. That the mechanism by which muscimol is acting on NINJ1 remains undefined is perhaps not surprising given how little we know about NINJ1 activation and regulation in plasma membrane rupture. This molecular understanding may require further integration with the recent (Degen M et al Nature 2023) and forthcoming structural studies. Nevertheless, by providing further insight on the pharmacologic inhibition of NINJ1, they complement the work of Borges JP et al (eLife 2022) and Kayagaki N et al (Nature 2023) by expanding the armamentarium by which NINJ1 can be targetted in cytoprotective strategies designed. As such, the findings will be of interest to the field of lytic cell death.

We thank the reviewer for their positive comments and support for our study.

One minor correction: Supplemental Fig 3 is missing a label for the y axis.

A label for the y axis has been added to revised Supplemental Fig 3.